# OBJECT-CENTRIC LEARNING WITH SLOT MIXTURE MODULE

**Daniil Kirilenko**[1,2]**, Vitaliy Vorobyov**[1]**, Alexey K. Kovalev**[1,3,4]**, Aleksandr I. Panov**[1,3,4]

[1] FRC CSC RAS, Moscow, Russia
[2] Università della Svizzera italiana, Lugano, Switzerland
[3] AIRI, Moscow, Russia
[4] MIPT, Dolgoprudny, Russia
`daniil.kirilenko@usi.ch, kovalev@airi.net`

## ABSTRACT

Object-centric architectures usually apply a differentiable module to the entire feature map to decompose it into sets of entity representations called slots. Some of these methods structurally resemble clustering algorithms, where the cluster's center in latent space serves as a slot representation. Slot Attention is an example of such a method, acting as a learnable analog of the soft k-means algorithm. Our work employs a learnable clustering method based on the Gaussian Mixture Model. Unlike other approaches, we represent slots not only as centers of clusters but also incorporate information about the distance between clusters and assigned vectors, leading to more expressive slot representations. Our experiments demonstrate that using this approach instead of Slot Attention improves performance in object-centric scenarios, achieving state-of-the-art results in the set property prediction task.

## 1 INTRODUCTION

In recent years, interest in object-centric representations has significantly increased (Greff et al., 2019; Burgess et al., 2019; Li et al., 2020; Engelcke et al., 2019; 2021; Locatello et al., 2020; Seitzer et al., 2022; Chakravarthy et al., 2023; Kirilenko et al., 2023). Such representations have the potential to improve the generalization ability of machine learning methods in many domains, such as reinforcement learning (Keramati et al., 2018; Watters et al., 2019a; Kulkarni et al., 2019; Berner et al., 2019; Sun et al., 2019), scene representation and generation (El-Nouby et al., 2019; Matsumori et al., 2021; Kulkarni et al., 2019; Kirilenko et al., 2022), reasoning (Yang et al., 2020; Kovalev et al., 2022), object-centric visual tasks (Groth et al., 2018; Yi et al., 2020; Singh et al., 2022), and planning (Migimatsu & Bohg, 2020).

Automatic segmentation of objects in the scene and the formation of a structured latent space can be achieved in various ways (Greff et al., 2020): augmenting features with grouping information, using a tensor product representation, incorporating ideas from attractor dynamics, etc. However, the most common and effective method for learning object-centric representations is Slot Attention (Locatello et al., 2020). Slot Attention maps the input feature vector received from the convolutional encoder to a fixed number of output feature vectors, referred to as slots. Through training, each object is assigned a corresponding slot. If the number of slots exceeds the number of objects, then some slots remain empty (i.e., they do not contain objects). This approach has shown significant results in object-centric tasks like set property prediction and object discovery. The limitations of this model are that significant object discovery results are only achieved on simple synthetic data, such as CLEVR (Johnson et al., 2017), Tetrominos, and Multi-dSprites (Kabra et al., 2019). On complex synthetic datasets such as ClevrTex (Karazija et al., 2021) or real data such as COCO (Lin et al., 2014), Slot Attention shows underperforming results. Also, on CLEVR, the standard dataset for the set property prediction task, Slot Attention, shows significant results only with weak thresholds, i.e., in cases where the model can be very mistaken regarding the position of objects. Another drawback of Slot Attention is its training instability, making it difficult to reproduce the results.

It is worth noting that most modern work evaluates object-centric methods mainly only on the object discovery task, i.e., recovering object masks, which can be seen as an unsupervised segmentation

problem. We believe that the results on the set property prediction task are also good indicators for object-centric methods, since this task requires distinguishing objects from each other and assigning them to a given set of objects.

In our work, we propose a generalization of the Slot Attention approach, replacing the k-means algorithm with a Gaussian Mixture Model (GMM) (Bauckhage, 2015), which we refer to as the Slot Mixture Module[1]. This leads to more expressive slot representations by incorporating information about the distance between clusters and their assigned vectors. Considering not only the weighted mean of a group of vectors as the representation gives an advantage in distinguishing similar but different groups. For example, two sets of vectors, sampled from different distributions but with the same expectation, would be indistinguishable if their mean were the only representation. This method enhances object-centric problem-solving across various tasks and datasets, surpassing even highly specialized models in the set property prediction task.

We conduct thorough experimental studies to substantiate the results obtained. Through extensive experiments, we show that the proposed Slot Mixture Module achieves the state-of-the-art performance in the set property prediction task on the CLEVR dataset (Johnson et al., 2017), outperforming even highly specialized models (Zhang et al., 2019). We provide experimental results for the image reconstruction task on five datasets: four with synthetic images (CLEVR-Mirror (Singh et al., 2022), ShapeStacks (Groth et al., 2018), ClevrTex (Karazija et al., 2021), Bitmoji (Graux, 2021)) and one with real-life images (COCO-2017 (Lin et al., 2014)). The proposed Slot Mixture Module consistently improves the performance of the GPT decoder. We show that the Slot Mixture Module outperforms the original Slot Attention on the object discovery task on such a complex dataset as ClevrTex. We also give an example of the use of advances in our approach, extending the idea of a concept library by estimating concept distributions. To justify the replacement of k-means by GMM clustering, we compare k-means and GMM clustering approaches on the set property prediction task and show that GMM clustering is a better choice for object-centric learning.

The main contributions of our paper are as follows: 1) We propose a generalization of the slot-based approach for object-centric representations, referred to as the Slot Mixture Module (Section 3); 2) Through extensive experiments, we demonstrate that our approach consistently improves performance in both unsupervised (image reconstruction in Section 4.1 and object discovery in Sections 4.3) and supervised (set property prediction in Section 4.2) object-oriented tasks across various achieving state-of-the-art results in the set property prediction task (Section 4.2) with stringent thresholds on the CLEVR dataset; 3) We demonstrate how additionally estimated slot mixture weights might aid in constructing a clear concept library by distinguishing between empty and filled slots. We also extend this approach to concept sampling (Section 4.4).

## 2 BACKGROUND

### 2.1 SLOT ATTENTION

Slot Attention (SA) module (Locatello et al., 2020) is an iterative attention mechanism designed to map a distributed feature map $\mathbf{x} \in \mathbb{R}^{N \times D}$ to a set of $K$ slots $\mathbf{s} \in \mathbb{R}^{K \times D}$. Slots are initialized randomly, and a trainable matrix $q \in \mathbb{R}^{D \times D}$ is used to get query projections of the slots, while $k, v \in \mathbb{R}^{D \times D}$ matricies are used to get key and value vectors from the feature map $\mathbf{x}$. Dot-product attention (Luong et al., 2015) between $q$ and $k$ projections with SoftMax across the $q$ dimension implies competition between slots for explaining parts of the input. Attention coefficients $A \in \mathbb{R}^{N \times K}$ are used to assign $v$ projections to slots via a weighted mean.

$$M = \frac{1}{\sqrt{D}} k(\mathbf{x}) q(\mathbf{s})^T \in \mathbb{R}^{N \times K}, \quad A_{i,j} = \frac{e^{M_{i,j}}}{\sum_{j=1}^{K} e^{M_{i,j}}}, \tag{1}$$

$$W_{i,j} = \frac{\mathbf{A}_{i,j}}{\sum_{i=1}^{N} \mathbf{A}_{i,j}}, \quad \mathbf{s}^* = W^T v(\mathbf{x}) \in \mathbb{R}^{K \times D}. \tag{2}$$

The Gated Recurrent Unit (GRU) (Cho et al., 2014) is employed for slot refinement, taking pre-updated slot representations $\mathbf{s}$ as hidden states and the updated slots $\mathbf{s}^*$ as inputs. A key characteristic of SA is its permutation invariance to input vectors and permutation equivariance for slots. This makes SA apt for handling sets and object-centric representations.

---

[1]The code is available at `https://github.com/AIRI-Institute/smm`

Technically, Slot Attention is a learnable analog of the k-means clustering algorithm with an additional trainable GRU update step and dot product (with trainable $q$, $k$, $v$) projections instead of Euclidean distance as the measure of similarity between the input vectors and cluster centroids. At the same time, k-means clustering can be considered as a special case of the Gaussian Mixture Model.

## 2.2 MIXTURE MODELS

Mixture Models (MM) is a class of parametric probabilistic models, in which it is assumed that each $\boldsymbol{x}_i$ from some observations $\boldsymbol{X} = \{\boldsymbol{x}_1, ..., \boldsymbol{x}_N\} \in \mathbb{R}^{N \times D}$ is sampled from the mixture distribution with $K$ mixture components and prior mixture weights $\boldsymbol{\pi} \in \mathbb{R}^K$:

$$\boldsymbol{x}_i \sim p(\boldsymbol{x}_i|\boldsymbol{\theta}) = \sum_{k=1}^K \pi_k p(\boldsymbol{x}_i|\boldsymbol{\theta}_k), \quad P(\boldsymbol{X}|\boldsymbol{\theta}) = \prod_{i=1}^N p(\boldsymbol{x}_i|\boldsymbol{\theta}), \quad \sum_k \pi_k = 1. \tag{3}$$

These models can be seen as the models with latent variables $z_{i,k} \in \{z_1, ..., z_K\}$ that indicate which component $\boldsymbol{x}_i$ came from. The problem is to find such $K$ groups of component parameters $\boldsymbol{\theta}_k$ and component assignments of each sample $\boldsymbol{x}_i$ that will maximize the likelihood of the model $P(\boldsymbol{X}|\boldsymbol{\theta})$. The Expectation Maximization (EM) algorithm is an iterative algorithm that addresses this problem and includes two general steps. The **Expectation (E) step**: evaluate the expected value of the complete likelihood $P(\boldsymbol{X}, \boldsymbol{Z}|\boldsymbol{\theta}^*)$ with respect to the conditional distribution of $P(\boldsymbol{Z}|\boldsymbol{X}, \boldsymbol{\theta})$:

$$Q(\boldsymbol{\theta}^*, \boldsymbol{\pi}^*|\boldsymbol{\theta}, \boldsymbol{\pi}) = \mathbb{E}_{P(\boldsymbol{Z}|\boldsymbol{X}, \boldsymbol{\theta})}[\log P(\boldsymbol{X}, \boldsymbol{Z}|\boldsymbol{\theta}^*)], \quad P(\boldsymbol{X}, \boldsymbol{Z}|\boldsymbol{\theta}^*) = \prod_{i=1}^N \prod_{k=1}^K [\pi_k p(\boldsymbol{x}_i|\boldsymbol{\theta}_k^*)]^{I(z_i = z_k)}, \tag{4}$$

where $I(*)$ is an indicator function.

The **Maximization (M) step**: find $\boldsymbol{\theta}^*, \boldsymbol{\pi}^*$ that maximize $Q(\boldsymbol{\theta}^*, \boldsymbol{\pi}^*|\boldsymbol{\theta}, \boldsymbol{\pi})$:

$$(\boldsymbol{\theta}, \boldsymbol{\pi}) = \text{argmax}_{(\boldsymbol{\theta}^*, \boldsymbol{\pi}^*)} Q(\boldsymbol{\theta}^*, \boldsymbol{\pi}^*|\boldsymbol{\theta}, \boldsymbol{\pi}). \tag{5}$$

One of the most widely used models of this kind is Gaussian Mixture Model (GMM), where each mixture component is modeled as a Gaussian distribution parameterized with its mean values and covariance matrix, which is diagonal in the simplest case: $P(\boldsymbol{x}_i|\boldsymbol{\theta}_k) = \mathcal{N}(\boldsymbol{x}_i|\boldsymbol{\mu}_k, \boldsymbol{\Sigma}_k)$, $\boldsymbol{\Sigma}_k = \text{diag}(\boldsymbol{\sigma}_k^2)$. In this case, the EM algorithm is reduced to the following calculations.

**E step**:
$$p(z_k|\boldsymbol{x}_i) = \frac{p(z_k)p(\boldsymbol{x}_i|\boldsymbol{\theta}_k)}{\sum_{k=1}^K p(z_k)p(\boldsymbol{x}_i|\boldsymbol{\theta}_k)} = \frac{\pi_k \mathcal{N}(\boldsymbol{x}_i|\boldsymbol{\mu}_k, \boldsymbol{\Sigma}_k)}{\sum_{k=1}^K \pi_k \mathcal{N}(\boldsymbol{x}_i|\boldsymbol{\mu}_k, \boldsymbol{\Sigma}_k)} = \gamma_{k,i}. \tag{6}$$

**M step**:

$$\pi_k^* = \frac{\sum_{i=1}^N \gamma_{k,i}}{N}, \quad \boldsymbol{\mu}_k^* = \frac{\sum_{i=1}^N \gamma_{k,i} \boldsymbol{x}_i}{\sum_{i=1}^N \gamma_{k,i}}, \quad \boldsymbol{\Sigma}_k^* = \frac{\sum_{i=1}^N \gamma_{k,i}(\boldsymbol{x}_i - \boldsymbol{\mu}_k^*)(\boldsymbol{x}_i - \boldsymbol{\mu}_k^*)^T}{\sum_{i=1}^N \gamma_{k,i}}. \tag{7}$$

The key difference between the Gaussian Mixture Model and k-means clustering is that GMM considers not only the centers of clusters but also the distance between clusters and assigned vectors with the prior probabilities of each cluster.

## 3 SLOT MIXTURE MODULE

For object-centric learning, we propose a modified Gaussian Mixture Model approach called the Slot Mixture Module (SMM). This module uses GMM **E** and **M steps** (Section 2.2) to map feature maps from the convolutional neural network (CNN) encoder to the set of slot representations, where slots are a concatenation of mean values and diagonal of the covariance matrix. Like Slot Attention, SMM uses GRU, which takes current and previous mean values as input and hidden states. In SA, slot representations are cluster centers, so SA is limited by the information contained and represented in these cluster centers. Considering not only the weighted mean of a group of vectors

---

**Algorithm 1** The Slot Mixture Module pseudocode. $\boldsymbol{\pi}$ is initialized as a uniform categorical distribution, $\boldsymbol{\mu}$ and $\boldsymbol{\Sigma}_{diag}$ are initialized from Gaussian distributions with trainable parameters.

---

**Input:** $\boldsymbol{x} \in \mathbb{R}^{N \times D}$ — flattened CNN feature map with added positional embeddings; $\boldsymbol{\mu}, \boldsymbol{\Sigma}_{diag} \in \mathbb{R}^{K \times D}, \boldsymbol{\pi} \in \mathbb{R}^{K}$ — SMM initialization parameters.
**Output:** slots $\in \mathbb{R}^{N \times 2D}$
$\boldsymbol{x} = \text{MLP}(\text{LayerNorm}(\boldsymbol{x}))$
**for** $t = 0...T$ **do**
    logits $= f_\theta(\boldsymbol{x}, \boldsymbol{\mu}, \boldsymbol{\Sigma}_{diag})$
    $\gamma = \text{SoftMax}(\text{logits} + \log \boldsymbol{\pi}, \text{dim}=K)$
    $\boldsymbol{\pi} = \gamma.\text{mean}(\text{dim}=N)$
    $\boldsymbol{\mu}^* = \text{WeightedMean}(\text{weights}=\gamma, \text{values}=\boldsymbol{x})$
    $\boldsymbol{\mu} = \text{GRU}(\text{input}=\boldsymbol{\mu}^*, \text{hidden}=\boldsymbol{\mu})$
    $\boldsymbol{\mu} \mathrel{+}= \text{MLP}(\text{LayerNorm}(\boldsymbol{\mu}))$
    $\boldsymbol{\Sigma}_{diag} = \text{WeightedMean}(\text{weights}=\gamma, \text{values}=(\boldsymbol{x} - \boldsymbol{\mu})^2))$
**end for**
slots $= \text{concat}([\boldsymbol{\mu}, \boldsymbol{\Sigma}_{diag}], \text{dim}=D)$
**return** slots

---

as the representation gives an advantage in distinguishing similar but different groups. For example, two sets of vectors, sampled from different distributions but with the same expectation, would be indistinguishable if their mean were the only representation. This set of slots is further used in the downstream task. We also use the same additional neural network update step for the mean values before updating the covariance values:

$$\boldsymbol{\mu}_k = \text{RNN}(\text{input}=\boldsymbol{\mu}_k^*, \text{hidden}=\boldsymbol{\mu}_k), \quad \boldsymbol{\mu}_k = \text{MLP}(\text{LayerNorm}(\boldsymbol{\mu}_k)) + \boldsymbol{\mu}_k. \tag{8}$$

These two steps serve the needs of the downstream task by linking the external and internal models. The internal model (E and M steps in SMM) tries to update its parameters $\boldsymbol{\mu}, \boldsymbol{\Sigma}$ so that the input vectors $x$ are assigned to slots with the maximum likelihood. In contrast, the external model takes these parameters as input. The Algorithm 1 presents the full pseudocode. A function $f_\theta(\boldsymbol{x}, \boldsymbol{\mu}, \boldsymbol{\Sigma}_{diag}) : \mathbb{R}^{N \times D} \times \mathbb{R}^{K \times D} \to \mathbb{R}^{N \times K}$ stands for a log-Gaussian density function which results in a matrix of log-probabilities, where each element reflects the estimated log-probability of a vector from $\boldsymbol{x}$ being generated by a corresponding Gaussian distribution.

It should be noted that SMM does not perform EM updates explicitly because the algorithm uses neural network updates that are not trained to maximize expectations as in traditional EM. Instead, SMM combines key update steps from the classical EM algorithm with trainable updates specific to neural networks. By incorporating trainable updates into the algorithmic cycle, it leverages the principles and insights of the EM algorithm while taking advantage of the flexibility and adaptability of neural networks. Combining classic EM steps with trainable updates allows for a more dynamic and data-driven approach, potentially leading to improved performance and adaptability to different scenarios.

The Slot Mixture Module can be seen as an extension of the Slot Attention Module (Figure 1) with the following key differences: (1) SMM updates not only the mean values but also the covariance values and prior probabilities, (2) the Gaussian density function is used instead of the dot-product attention, and (3) slots are considered not only as mean values of the cluster but as the concatenation of mean and covariance values.

## 4 EXPERIMENTS

Since our model can be seen as an extension of the Slot Attention, we use it as our main competitor. Our approach predicts the mean $\boldsymbol{\mu}$ and covariance $\boldsymbol{\Sigma}_{diag}$, which doubles the intermediate slot representation, so we apply an additional matrix to effectively reduce the dimensionality of the intermediate slot representations by a factor of two (the matrix size is $2D \times D$). Thus, the final size of the slots produced by SMM is the same as that of the slots produced by SA. It is also important to note that both SMM and SA have the same number of trainable parameters because SMM does not use $k$ and $v$ projections (the size of the $k$ and $v$ matrices is $D \times D$), which ensures a fair comparison between the two approaches.

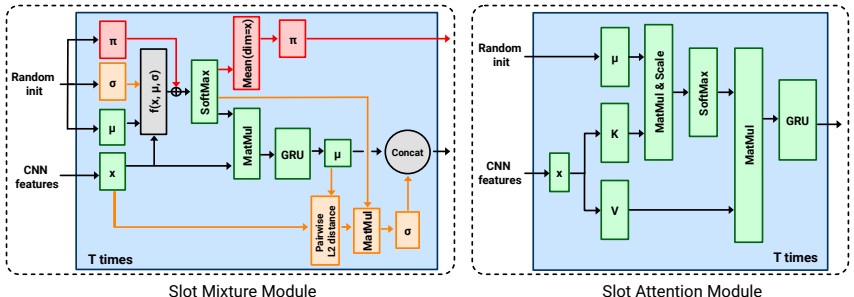

Figure 1: Visualized architectures of Slot Mixture Module (ours) and Slot Attention Module. Green color is used for steps involved in both modules. SMM involves the estimation of cluster centers ($\boldsymbol{\mu}$), the distance between cluster centers and assigned vectors ($\boldsymbol{\sigma}$, orange steps), and prior mixture weights ($\pi$, red steps). The concatenation of $\boldsymbol{\mu}$ and $\boldsymbol{\sigma}$ serves as slot representations, and $\pi$ can be used to identify empty slots that do not contain information about any objects, $f(\boldsymbol{x}, \boldsymbol{\mu}, \boldsymbol{\sigma})$ is the log of the Gaussian PDF. SA module estimates only centers of clusters.

In experiments, we trained two versions of the same model: the one with the Slot Attention module and another with our SMM, keeping the same training conditions. We use the technique of detaching slots from the gradient computational graph at the last iteration of the algorithm from Chang et al. (2022) to achieve stable training and avoid exploding gradients. This work also shows that training with a small number of iterations (1 iteration) can lead to significantly worse performance in specific tasks than training with a larger number of iterations (7 iterations). We use five iterations as a good default choice in the trade-off between best performance and excessive computation. Since the optimal number of steps may depend on the datasets and tasks, we did not perform a hyperparameter search, as this would lead to difficulties in comparing different tasks and datasets. We choose the average number of 5 based on the most related works: Locatello et al. (2020) uses 3, Singh et al. (2022) uses 7 for some datasets and 3 for others, Chang et al. (2022) shows the difference in performance between 1 and 7 iterations. Additional architectural details are presented in Appendix A.

## 4.1 IMAGE RECONSTRUCTION USING TRANSFORMER

We use the SLATE model (Singh et al., 2022), substituting SA with our SMM for an unsupervised image-to-image task comparison. SLATE employs an Image GPT (Chen et al., 2020) decoder conditioned on slot representations to autoregressively reconstruct discrete visual tokens from a discrete VAE (dVAE) (Im et al., 2017). It treats pre-computed slot representations as query vectors and image latent code vectors as key/value vectors. Although SLATE excels at capturing complex slot-pixel interactions in synthetic images, its performance wanes with real-world data. The training signals for the dVAE encoder, decoder, and latent discrete tokens are derived from the MSE between the input and reconstructed images. SA/SMM modules and Image GPT are trained with cross-entropy using the compressed image into dVAE tokens as the target distribution. This process is isolated from the rest of the model (i.e., dVAE), but both systems are trained simultaneously.

**Setup.** We consider the following datasets: CLEVR-Mirror (Singh et al., 2022), ClevrTex (Karazija et al., 2021), ShapeStacks (Groth et al., 2018), and COCO-2017 (Lin et al., 2014). CLEVR-Mirror is an extension of the standard CLEVR dataset, which requires capturing global relations between local components due to the presence of a mirror. ShapeStacks tests the ability of the model to describe complex local interactions (multiple objects stacked on each other), and ClevrTex examines the model's capabilities in textural-rich scenes. For ShapeStacks, ClevrTex, and COCO, we used images rescaled to the resolution of $96 \times 96$, and CLEVR-Mirror images are rescaled to $64 \times 64$. Training conditions with hyperparameters corresponding to a certain dataset are taken from Singh et al. (2022), except that we use a batch size equal to $64$ and $2.5 \times 10^5$ training iterations for all experiments.

**Results.** Table 1 shows the metrics of the image reconstruction performance for the test parts of different datasets. Frechet Inception Distance (FID) (Heusel et al., 2017) computed with the PyTorch-Ignite library (Fomin et al., 2020) as a measure of the quality of the generated images. We also evaluated cross-entropy between tokens from dVAE (as one-hot distributions) and predicted distributions by Image GPT that are conditioned by certain slot representations, as FID can be limited by the discrete VAE. SLATE has two parts that are trained separately: the dVAE and the autoregressive transformer. The transformer takes slot representations as input and aims to reconstruct

Table 1: Reconstruction performance (mean ± std for 4 seeds). SLATE(SA) — the original SLATE model, SLATE(SMM) — modified SLATE with SMM instead of SA. For ceiling performance, FID results for dVAE are provided.

| | FID | | | CROSS-ENTROPY | |
| DATA | SLATE(SA) | SLATE(SMM) | DVAE | SLATE(SA) | SLATE(SMM) |
| --- | --- | --- | --- | --- | --- |
| CLEVR-MIRROR | 35.4 ± 1.1 | 34.8 ± 0.9 | 32.2 ± 0.9 | 0.82 ± 0.05 | **0.20 ± 0.02** |
| SHAPESTACKS | 56.6 ± 2.8 | **50.4 ± 1.84** | 40.4 ± 1.3 | 88.3 ± 2.2 | 62 ± 2.1 |
| CLEVRTEX | 116 ± 4.6 | 113 ± 4.3 | 82.8 ± 3.9 | 566 ± 18.1 | 507 ± 21.7 |
| COCO | 129 ± 5.2 | 122 ± 4.1 | 92.0 ± 4.4 | 540 ± 17.5 | 461 ± 12.1 |
| CELEBA | 39.2 ± 1.5 | 41.3 ± 1.6 | 35.7 ± 1.1 | 734 ± 31.9 | 655 ± 28.6 |
| BITMOJI | 30.8 ± 0.52 | **27.0 ± 0.57** | 25.1 ± 0.4 | 15.1 ± 0.16 | **13.3 ± 0.21** |

tokens from the dVAE encoder by minimizing the cross-entropy between ground truth tokens and predicted tokens. Therefore, we consider lower cross-entropy as evidence of higher expressiveness of SMM representations compared to SA representations since it helps to reconstruct original tokens accurately. The better prediction of dVAE-encoded tokens leads to better image reconstruction performance, even though it was not directly trained for this since only the dVAE was trained to minimize image reconstruction loss.

In Table 1, we report results for the SLATE model and its modification with SMM. SLATE does not output masks like an object discovery model from the SA paper (Locatello et al., 2020) because SLATE uses attention maps as a heuristic to visualize each slot's information. It would be incorrect to treat them as masks for computing the FG-ARI since the task is not object discovery.

In Appendix B, we provide validation cross-entropy curves during training. Figures 2 and 3 display reconstructed images from ShapeStacks, ClevrTex, and CLEVR-Mirror datasets, highlighting SMM's ability to recreate the correct object order from original images more accurately.

**ShapeStacks**      **ClevrTex**      **CLEVR-Mirror**

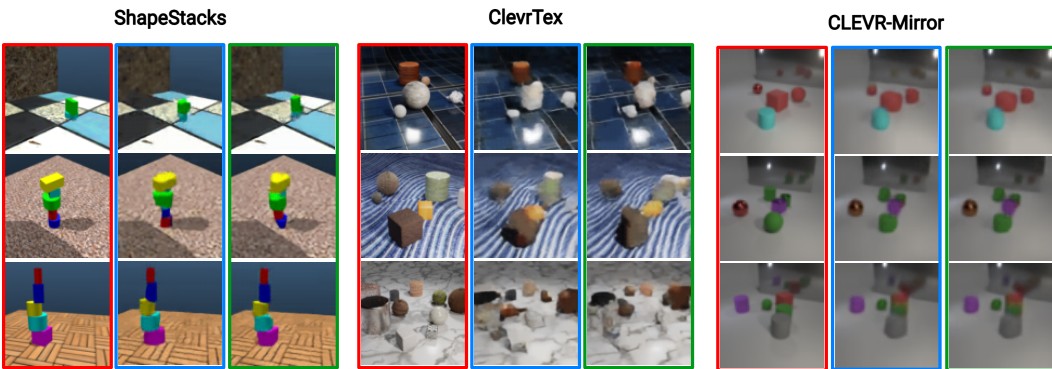

Figure 2: Examples of image generation with Image GPT conditioned to different slot representations. Images in the blue borders are from the model with the Slot Attention module, and images in green borders are generated using slots from the Slot Mixture Module. Red color stands for input images.

### 4.2 SET PROPERTY PREDICTION

Neural networks for sets are involved in various applications across many data modalities (Carion et al., 2020; Achlioptas et al., 2017; Simonovsky & Komodakis, 2018; Fujita et al., 2019). Set Property Prediction is a supervised task that requires the model to predict an unordered set of vectors representing the properties of objects from the input image. Sets of predicted and target vectors are matched using a Hungarian algorithm (Kuhn, 1955) and the learning signal is provided by Huber Loss (Zhang et al., 2019) between matched vectors. The Slot Mixture Module is suitable for operating with sets as it preserves permutation equivariance regarding mixture components (slots) and initializes them randomly.

**Setup.** We use the CLEVR dataset because it is standard for comparing object-centric models on the set property prediction task. We rescaled images to a resolution of $128 \times 128$ as the data source.

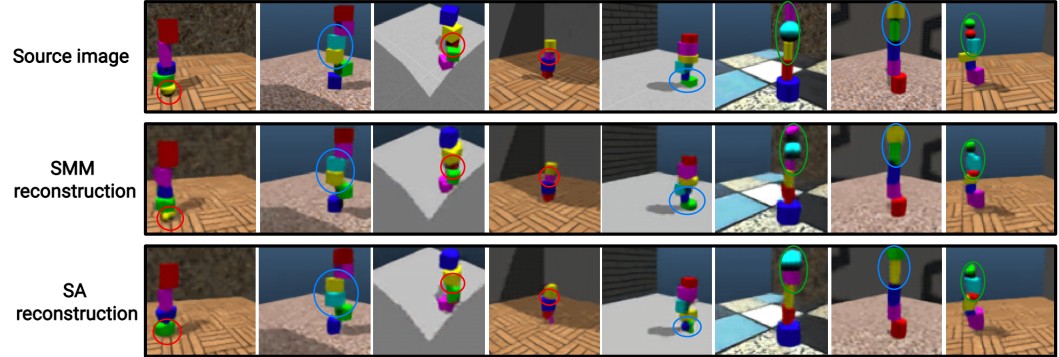

Figure 3: Examples of all the qualitatively incorrectly generated images from the random batch of 64 samples. In 6 cases, reconstruction using Slot Attention gave a wrong order of objects (blue circle) or lost one object (red circle). In the remaining two samples, both modules gave incorrect reconstruction (green circle).

All modules except SA/SMM are the same as in the (Locatello et al., 2020). Each model is trained with Adam optimizer (Kingma & Ba, 2015) for $1.5 \times 10^5$ iterations with OneCycleLR (Smith & Topin, 2019) learning rate scheduler at a maximum learning rate of $4 \times 10^{-4}$. We use a batch size of 512. SA/SMM number of iterations is set to 5 during training and 7 during evaluation. The number of slots equals 10 since CLEVR images contain 10 or fewer objects.

**Results.** Table 2 showcases the quantitative results of our experiments, where we compute Average Precision (AP) using varied distance thresholds for object property and coordinate prediction. Our findings indicate that modifications to the original Slot Attention — such as detaching slots before the final refinement iteration and rescaling coordinates — can enhance its performance. However, it still trails behind the state-of-the-art iDSPN model. With non-strict thresholds, other SA modifications (SA-MESH (Zhang et al., 2023)) also produce near-ideal results. In contrast, SMM consistently outperforms a highly specialized model such as iDSPN and achieves state-of-the-art results with stringent thresholds (starting at 0.25).

Table 2: Set prediction performance on the CLEVR dataset (AP in %, mean ± std for 4 seeds in our experiments). Slot Attention (Locatello et al., 2020) and iDSPN (Zhang et al., 2019) results are from the original papers. SA* is the Slot Attention model trained with the same conditions as our SMM: detached slots at the last iteration rescaled coordinates to the range of [-1, 1]. SA-MESH results are taken from the original paper (Zhang et al., 2023).

| MODEL | $AP_\infty$ | $AP_1$ | $AP_{0.5}$ | $AP_{0.25}$ | $AP_{0.125}$ | $AP_{0.0625}$ |
|---|---|---|---|---|---|---|
| SA | $94.3 \pm 1.1$ | $86.7 \pm 1.4$ | $56.0 \pm 3.$ | $10.8 \pm 1.7$ | $0.9 \pm 0.2$ | – |
| SA* | $97.1 \pm 0.7$ | $94.5 \pm 0.7$ | $88.3 \pm 3.2$ | $62.5 \pm 5.4$ | $23.6 \pm 1.4$ | $4.6 \pm 0.3$ |
| SA-MESH | $99.4 \pm 0.1$ | $99.2 \pm 0.2$ | $98.9 \pm 0.2$ | $91.1 \pm 1.1$ | $47.6 \pm 0.8$ | $12.5 \pm 0.4$ |
| iDSPN | $98.8 \pm 0.5$ | $98.5 \pm 0.6$ | $98.2 \pm 0.6$ | $95.8 \pm 0.7$ | $76.9 \pm 2.5$ | $32.3 \pm 3.9$ |
| SMM (OURS) | $99.4 \pm 0.2$ | $99.3 \pm 0.2$ | $98.8 \pm 0.4$ | $\mathbf{98.4 \pm 0.7}$ | $\mathbf{92.1 \pm 1.2}$ | $\mathbf{47.3 \pm 2.5}$ |

### 4.3 OBJECT DISCOVERY

Another unsupervised object-centric image-to-image task is Object Discovery. We use a model from Locatello et al. (2020) for this task. This approach decodes each slot representation into the 4-channel image using a Spatial Broadcast decoder (Watters et al., 2019b). The resulting reconstruction in the pixel space is estimated as a mixture of decoded slots, where the first three channels are responsible for the reconstructed RGB image, and the fourth channel is for the weights of the mixture component that serve as masks.

**Setup.** Since SA shows almost perfect metrics on such simple datasets as CLEVR, we conducted experiments on a complex dataset — ClevrTex. We consider the same training setup from the original work Locatello et al. (2020) as for CLEVR, and the only difference is the ClevrTex image resolution is reduced to 64x64.

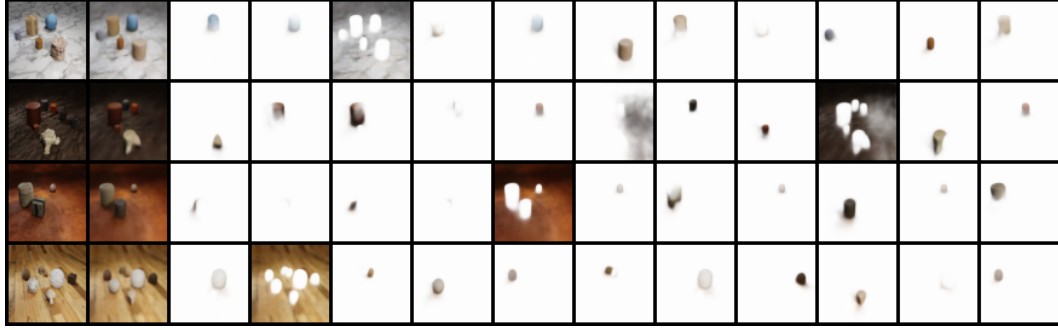

Figure 4: Example of object-discovery on ClevrTex dataset. The first column represents ground-truth images. The second one is broadcast-decoder reconstructions. The next columns are per slots of attention masks.

**Results.** Table 3 shows the similarity between ground truth segmentation masks of objects (excluded background) and mixture coefficients estimated via the Foreground Adjusted Rand Index (FG-ARI) score on ClevrTex (mean $\pm$ std for 3 seeds are reported). SMM outperforms SA on such a complex dataset. The examples of SMM object-discovery is shown in Figure 4. Additional results for SMM and SA are shown in Appendix C in Figure 7.

We have also conducted experiments on such simple datasets as CLEVR10, Multi-dSprites, and Tetrominoes. Since SA shows almost perfect results with long enough training on these datasets, we reduced the training steps to 300k (instead of 500k) to com-

Table 3: FG-ARI score for ClevrTex.

| Dataset | SA | SMM |
| --- | --- | --- |
| ClevrTex | $62.40 \pm 2.33$ | $\mathbf{74.7 \pm 1.3}$ |

pare the convergence of the approaches. The FG-ARI results are: i) CLEVR10 — SA $85.6 \pm 1.2$, SMM $91.3 \pm 0.6$; ii) Multi-dSprites — SA $81.0 \pm 0.7$, SMM $89.6 \pm 0.6$; iii) Tetrominoes —- SA $75.4 \pm 1.0$, SMM $83.2 \pm 0.7$. Our method shows faster convergence, resulting in better object segmentation.

## 4.4 CONCEPT SAMPLING

Building on the concept library approach introduced in Singh et al. (2022), we propose an extension that enhances slot collection by discarding empty slots. This refinement is possible thanks to the SMM's evaluation of prior mixture weights $\pi$. We then apply GMM clustering on this streamlined set, which retains the ability to assign slots to clusters, identify underlying concepts, and enable concept sampling due to the estimation of Gaussian distribution parameters.

Our approach to assessing image editing quality involves extracting and sampling new slots for each image from the validation dataset using the Gaussian parameters of their assigned clusters. These sampled slots are then input to the decoder to generate the resulting image. Experiments on Bitmoji, ShapeStacks, and CLEVR-Mirror datasets yielded FID scores of 32.1, 59, and 41.4, respectively, indicating a high quality of edited images comparable to regular autoencoding (Table 1). Figure 5 illustrates this concept sampling applied to a single slot.

## 4.5 COMPARING VANILLA CLUSTERING

Slot Attention and Slot Mixture modules can be reduced to k-means and Gaussian Mixture Model clustering approaches by removing GRU/MLP updates, trainable $q, k, v$ projections, and LayerNorm layers (Ba et al., 2016). Table 4 shows the training set property prediction model results for the CLEVR dataset using these vanilla clustering methods. Our experiments demonstrate that GMM clustering is better for object-centric learning, even without trainable layers. The results of the ablation studies are presented in the Appendix D.

Table 4: Average Precision (mean $\pm$ std for four seeds) with different distance thresholds for set property prediction task on the CLEVR dataset after 100k training iterations.

| MODEL | $AP_\infty$ | $AP_1$ | $AP_{0.5}$ | $AP_{0.25}$ | $AP_{0.125}$ |
| --- | --- | --- | --- | --- | --- |
| K-MEANS | $81.7 \pm 1.6$ | $49.1 \pm 2.9$ | $7.2 \pm 1.7$ | $1.4 \pm 0.3$ | $0.2 \pm 0.1$ |
| GMM | $\mathbf{88.6 \pm 2.0}$ | $\mathbf{53.3 \pm 2.2}$ | $\mathbf{9.2 \pm 1.6}$ | $\mathbf{2.3 \pm 0.3}$ | $\mathbf{0.5 \pm 0.1}$ |

## 5 RELATED WORKS

**Object-centric representation.** Object-centric learning involves the unsupervised identification of objects in a scene, a critical element of representation learning. The domain has evolved, starting with earlier methods that capitalized on sequential attention mechanisms such as AIR (Eslami et al., 2016) and its successors SQAIR (Kosiorek et al., 2018) and R-SQAIR (Stanić & Schmidhuber, 2019). The field also witnessed the advent of spatial mixture model-based methods like Tagger (Greff et al., 2016) and NEM (Greff et al., 2017), which are aimed to optimize the complete likelihood of the mixture model at the pixel level, with the latter focusing on unsupervised learning and perceptual grouping. R-NEM (Van Steenkiste et al., 2018) improved upon NEM by incorporating the discovery of object interactions. Recent advancements have integrated these principles, resulting in more sophisticated approaches like IODINE (Greff et al., 2019), MONET (Burgess et al., 2019), and GENESIS (Engelcke et al., 2019; 2021) that leverage the VAE framework (Kingma & Welling, 2014; Rezende et al., 2014). While these methods employ multi-step encoding and decoding, hybrid models like SPACE (Lin et al., 2020) utilize spatial attention for scene decomposition. Despite the diversity of approaches, a common characteristic is the move towards more efficient and effective representation of complex scenes. Sequential extensions to SA (Kipf et al., 2021; Elsayed et al., 2022) work with video data, while Biza et al. (2023) introduced Invariant Slot Attention, enhancing object representation invariance to geometric transformations, and Chakravarthy et al. (2023) integrated a spatial-locality prior into object-centric models. Seitzer et al. (2022) introduces DINOSAUR that leverages self-supervised learning to further enhance SA efficacy in real-world data scenarios. Recent advancements such as Hénaff et al. (2022) and Wang et al. (2023), while not directly related to learning object representations, have made significant contributions to the related field of unsupervised object segmentation.

**Set prediction.** Set-predicting neural network models are applied in diverse machine learning tasks (Achlioptas et al., 2017; Carion et al., 2020; Fujita et al., 2019; Simonovsky & Komodakis, 2018). Despite their practicality, traditional models often struggle with set representation. However, approaches like Deep Set Prediction Network (DSPN) and its improved version iDSPN (Zhang et al., 2019) account for sets' unordered nature through an inner gradient descent loop and approximate implicit differentiation, respectively. Set-equivariant self-attention layers are also utilized in models like TSPN for set structure representation (Kosiorek et al., 2020).

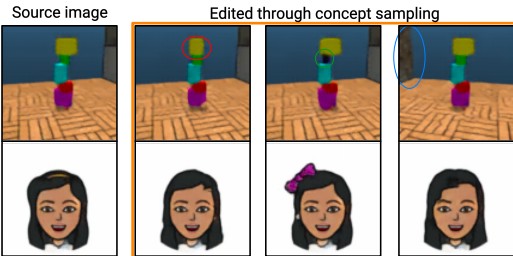

Figure 5: Example of editing images from ShapeStacks and Bitmoji using concept sampling. This method allows precise editing of individual concepts: the first row demonstrates the capability of changing shape, color of single objects or manipulating background. The second row gives an example of achieving similar yet slightly different hairstyles with concept sampling.

## 6 CONCLUSION AND DISCUSSION

This paper presents the Slot Mixture Module, a novel slot-based approach to object-centric representations that generalizes the Gaussian Mixture Model (Bauckhage, 2015). Our approach uniquely considers cluster centers and inter-cluster distances to enhance slot representations. Empirical evidence from the CLEVR dataset (Johnson et al., 2017) confirms our model's state-of-the-art performance in the set property prediction task with stringent thresholds and superior performance in the image reconstruction task. Furthermore, we demonstrate improved reconstruction performance on synthetic datasets like CLEVR-Mirror (Singh et al., 2022), ShapeStacks (Groth et al., 2018), and ClevrTex Karazija et al. (2021). SMM also outperoems SA on suach a complex dataset as ClevrTex on th object discovery task. However, the limited effectiveness of modern models (Seitzer et al., 2022) on real-life images, like COCO-17 (Lin et al., 2014), underscores the need for improved generalization to real data. We discuss limitations of the proposed approach in Appendix F.

**Reproducibility Statement** To improve the reproducibility of the results, we provide a detailed description of the method with the algorithm in pseudocode in Section 3, we provide all necessary technical details in Section 4 and Appendix A, we use only open datasets in the experiments, and we make the source code of our model publicly available at `https://github.com/AIRI-Institute/smm`.

ACKNOWLEDGMENTS

This work was supported by Russian Science Foundation, grant No. 20-71-10116, `https://rscf.ru/en/project/20-71-10116/`.

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

# A  ARCHITECTURE DETAILS

Tables 5, 6, 7, 8 describe hyperparameters for our experiments. In the case of using SMM instead of SA, we use an additional dimensionality reduction for slots via trainable matrix multiplication.

Table 5: Architecture of the CNN encoder for set property prediction and object discovery tasks experiments. The set prediction model uses a stride of 2 in the layers with *, while the object discovery model uses a stride of 1 in these layers.

| Layer | Channels | Activation | Params |
|---|---|---|---|
| Conv2D $5 \times 5$ | 64 | ReLU | stride: 1 |
| Conv2D $5 \times 5$ | 64 | ReLU | stride: 1/2* |
| Conv2D $5 \times 5$ | 64 | ReLU | stride: 1/2* |
| Conv2D $5 \times 5$ | 64 | ReLU | stride: 1 |
| Position Embedding | - | - | absolute |
| Flatten | - | - | dims: w, h |
| LayerNorm | - | - | - |
| Linear | 64 | ReLU | - |
| Linear | 64 | - | - |

Table 6: Spatial broadcast decoder for object discovery task on the CLEVR and ClevrTex datasets.

| Layer | Channels/Size | Activation | Params |
|---|---|---|---|
| Spatial Broadcast | $8 \times 8$ | - | - |
| Position Embedding | - | - | absolute |
| ConvTranspose2D $5 \times 5$ | 64 | ReLU | stride: 2 |
| ConvTranspose2D $5 \times 5$ | 64 | ReLU | stride: 2 |
| ConvTranspose2D $5 \times 5$ | 64 | ReLU | stride: 2 |
| ConvTranspose2D $5 \times 5$ | 64 | ReLU | stride: 2 |
| ConvTranspose2D $5 \times 5$ | 64 | ReLU | stride: 1 |
| ConvTranspose2D $3 \times 3$ | 4 | - | stride: 1 |
| Split Channels | RGB (3), mask (1) | Softmax on masks (slots dim) | - |
| Combine components | - | - | - |

Table 7: Spatial broadcast decoder for object discovery experiments on Tetrominoes and Multi-dSprites datasets.

| Layer | Channels/Size | Activation | Params |
|---|---|---|---|
| Spatial Broadcast | $64 \times 64$ | - | - |
| Position Embedding | - | - | absolute |
| ConvTranspose2D $5 \times 5$ | 32 | ReLU | stride: 1 |
| ConvTranspose2D $5 \times 5$ | 32 | ReLU | stride: 1 |
| ConvTranspose2D $5 \times 5$ | 32 | ReLU | stride: 1 |
| ConvTranspose2D $3 \times 3$ | 4 | - | stride: 1 |
| Split Channels | RGB (3), mask (1) | Softmax on masks (slots dim) | - |
| Combine components | - | - | - |

Table 8: Hyperparameters used for our experiments with the SLATE architecture.

| Module | Parameter | Value |
|---|---|---|
| | Image Size | 96 |
| | Encoded Tokens | 576 |
| dVAE | Vocab size | 4096 |
| dVAE | Temp. Cooldown | 1.0 to 0.1 |
| dVAE | Temp. Cooldown Steps | 30000 |
| dVAE | LR (no warmup) | 0.0003 |
| Transformer | Layers | 8 |
| Transformer | Heads | 8 |
| Transformer | Hidden Dim. | 192 |
| SA/SMM | Num. slots | 12 |
| SA/SMM | Iterations | 7 |
| SA/SMM | Slot dim. | 192 |

## B    ADDITIONAL RESULTS FOR IMAGE RECONSTRUCTION

Figure 6 shows validation cross-entropy curves during training.

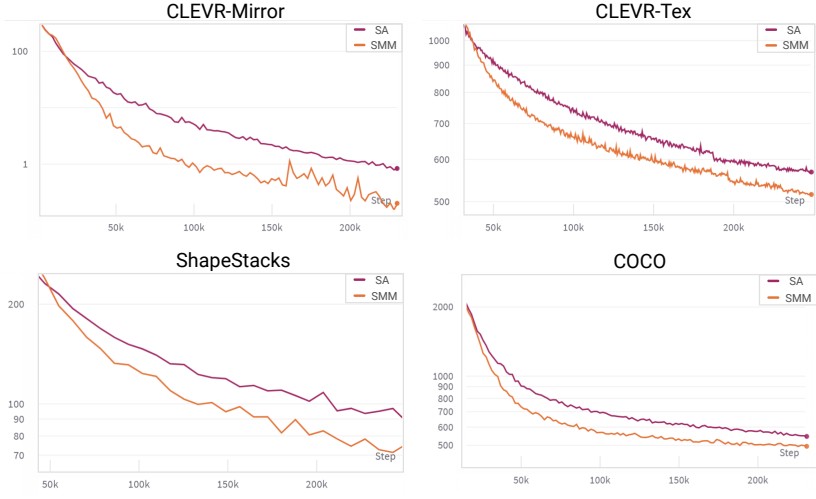

Figure 6: Validation cross-entropy during training for 4 different datasets. Our experiments show that using the SMM module instead of SA consistently improves the validation performance of the autoregressive transformer by about 10 percent during training. The result is maintained for all the datasets that we use.

## C    ADDITIONAL RESULTS FOR OBJECT DISCOVERY

Figure 7 shows examples of object-discovery on ClevrTex dataset for SMM and SA.

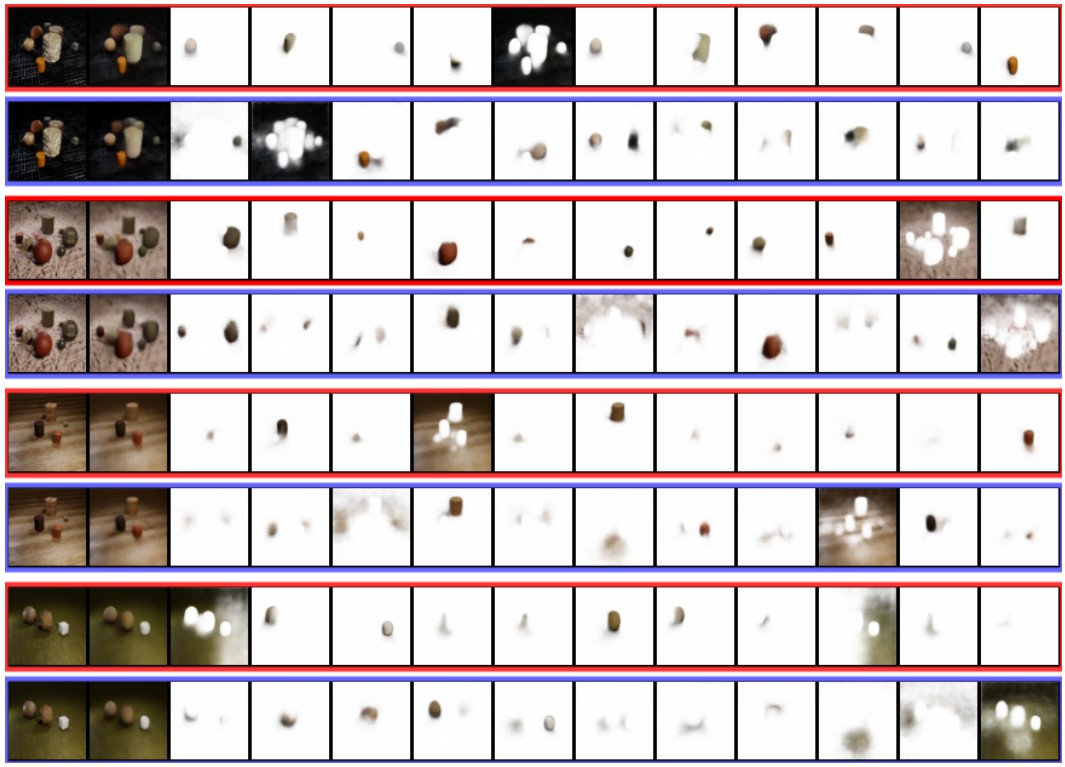

Figure 7: Examples of object-discovery on ClevrTex dataset for SMM and SA. The first column represents ground-truth images. The second one is broadcast-decoder reconstructions. The next columns are per slots of attention masks. Examples in the red borders are for SMM and examples in the blue borders are for SA.

## D    ABLATION STUDY

We also conducted the following ablation experiments. We trained Slot Attention (SA) and SMM object discovery models on the CLEVR6 dataset (scenes with six or fewer objects) with seven slots and three update iterations. Then we evaluated the trained models using different setups.

Evaluating the models on the CLEVR6 test split with 1, 2, ..., and 7 update iterations gives the following FG-ARI scores (see Table 9).

Table 9: FG-ARI score for different number of update iterations (from 1 to 7)) at test time on the CLEVR6 dataset.

| MODEL | 1 | 2 | 3 | 4 | 5 | 6 | 7 |
|-------|------|------|------|------|------|------|------|
| SA    | 68.0 | 90.1 | 99.3 | 99.5 | 99.5 | 99.6 | 99.5 |
| SMM   | 67.0 | 78.9 | 99.8 | 99.7 | 99.4 | 99.0 | 97.1 |

Evaluating the models on the CLEVR test images with only 7, 8, 9, and 10 objects (and a corresponding number of slots) gives the following results (see Table 10).

Table 10: FG-ARI score for different number of objects (from 7 to 10) at test time.

| MODEL | 7 | 8 | 9 | 10 |
|---|---|---|---|---|
| SA | 97.0 | 95.1 | 94.0 | 93.3 |
| SMM | 97.1 | 96.6 | 94.5 | 94.5 |

The results show that SMM is more robust to the number of out-of-distribution objects and less robust to different numbers of update iterations at test time.

## E HIGHER RESOLUTION OBJECT DISCOVERY

On Figure 8 and Figure 9are shown examples of object-discovery and image reconstruction on ClevrTex dataset with 128x128 resolution for SMM and SA. Model architecture is presented on Table11and Table12

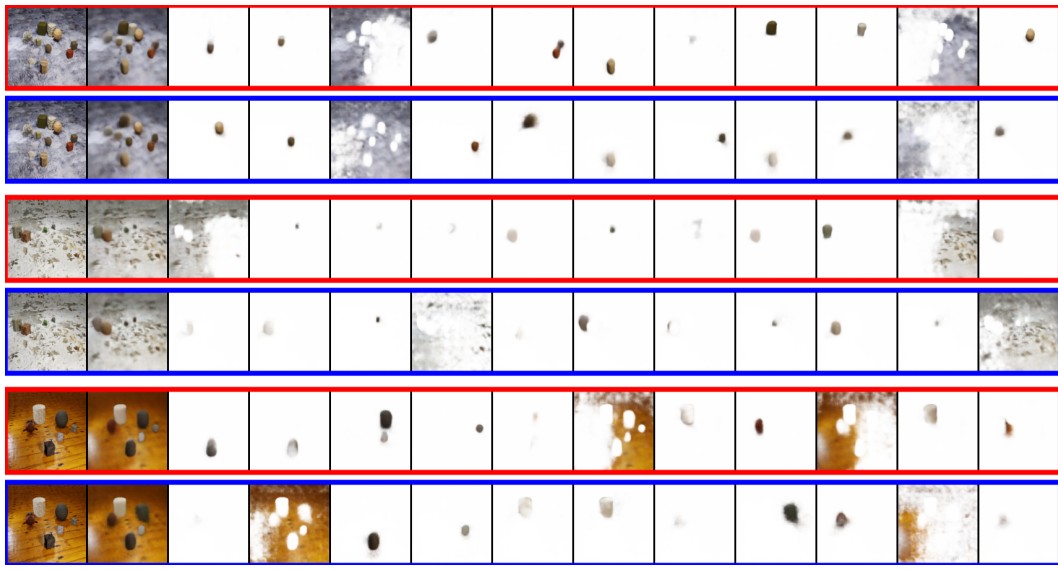

Figure 8: Examples of object-discovery on ClevrTex dataset with $128 \times 128$ image resolution for SMM and SA. The first column represents ground-truth images. The second one is broadcast-decoder reconstructions. The next columns are per slots of attention masks. Examples in the red borders are for SMM and examples in the blue borders are for SA.

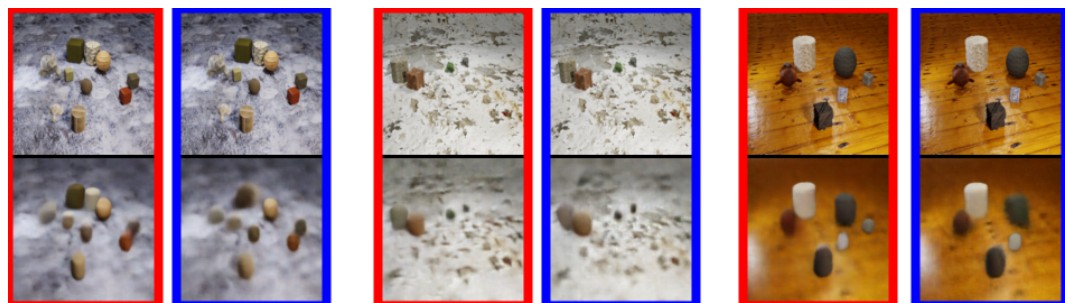

Figure 9: Image reconstructions on ClevrTex dataset with $128 \times 128$ resolution for SMM and SA. The first row represents ground-truth images. The second one is broadcast-decoder reconstructions. Examples in the red borders are for SMM and examples in the blue borders are for SA.

Table 11: Spatial broadcast decoder for object discovery task on the ClevrTex datasets with $128\times128$ image resolution.

| Layer | Channels/Size | Activation | Params |
|---|---|---|---|
| Spatial Broadcast | $8 \times 8$ | - | - |
| Position Embedding | - | - | absolute |
| ConvTranspose2D $5 \times 5$ | 64 | ReLU | stride: 2 |
| ConvTranspose2D $5 \times 5$ | 64 | ReLU | stride: 2 |
| ConvTranspose2D $5 \times 5$ | 64 | ReLU | stride: 2 |
| ConvTranspose2D $5 \times 5$ | 64 | ReLU | stride: 2 |
| ConvTranspose2D $5 \times 5$ | 64 | ReLU | stride: 1 |
| ConvTranspose2D $3 \times 3$ | 4 | - | stride: 1 |
| Split Channels | RGB (3), mask (1) | Softmax on masks (slots dim) | - |
| Combine components | - | - | - |

Table 12: Architecture of the CNN encoder for ClevrTex with image size $128 \times 128$.

| Layer | Channels | Activation | Params |
|---|---|---|---|
| Conv2D $5 \times 5$ | 64 | ReLU | stride: 2 |
| Conv2D $5 \times 5$ | 64 | ReLU | stride: 1 |
| Conv2D $5 \times 5$ | 64 | ReLU | stride: 1 |
| Conv2D $5 \times 5$ | 64 | ReLU | stride: 1 |
| Position Embedding | - | - | absolute |
| Flatten | - | - | dims: w, h |
| LayerNorm | - | - | - |
| Linear | 64 | ReLU | - |
| Linear | 64 | - | - |

## F LIMITATIONS

The main limitations of SMM are due to the fact that the model belongs to the class of slot models, since it is based on a prominent representative of this class - Slot Attention. These limitations are: the need to specify the number of slots in advance, thereby setting the maximum possible number of objects in the image; over/under segmentation based on the specified number of slots; as a result, poor quality with a large number of slots; poor results and not aligned with human perception segmentation of real data.

While we have enhanced real-world object-centric image generation, the overall quality remains subpar and the attention maps scarcely resemble human object-focused vision. Future research should address the challenge of scaling and refining these visual models for complex real-world images. Also, as shown in the ablation study, SMM is less robust to different numbers of update iterations at test time.

We also want to draw attention to the potential negative societal impact of object-centric models, since modification and replacement of objects represented by a slot in an image can be used for malicious purposes.

