# OpenReview forum: "Object-Centric Learning with Slot Mixture Module"
_ICLR.cc/2024/Conference — ICLR 2024 poster_

### Official Review · Reviewer_fszf · 2023-10-30

**Soundness:** 3 good
**Presentation:** 2 fair
**Contribution:** 2 fair
**Rating:** 5
**Confidence:** 4

**Summary:**

The paper studies object-centric learning based on the existing slot attention method. Particularly, it incorporates Gaussian mixture model (GMM) rather than the soft k-means clustering used in the original slot attention to learn better slot representations, achieving promising results on three tasks: image reconstruction, set prediction and object discovery.

**Strengths:**

1) The paper studies a very important problem of object-centric learning.

2) The introduced GMM mechanism to learn distinguishable slot representations appears to be more effective than using a single mean vector to represent each slot.

**Weaknesses:**

1) The key contribution of this paper is the application of GMM mechanism for better slot representation learning, but the improvements are more about the performance in supervised tasks: image reconstruction and set prediction, instead of the more desirable unsupervised object segmentation. This makes the general contribution to be less appealing.

2) In page 2, the claimed Contributions (2) and (3) are not very meaningful, or at least both can be combined, because both sentences describe the experimental results.

3) In Equations (1)(2), it's suggested to use math symbols instead of English words. It’s also suggested to use bold symbols to represent vectors.

4) In page 4, it is unclear how the function f_theta(x, u, diag) works. Obviously, the input has three elements instead of two as described in the text "R^{NxD} x R^{KxD} -> R^{NxK}". More details should be provided because this is the primary technique contribution of this paper.

5) In Section 4.3, for the experiments of object discovery, the dataset(ClevrTex) is a bit too simple and the evaluation metric ARI is actually not suitable as well because the scores can easily achieve perfect numbers. It is advised to evaluate on more complex (real-world) datasets and use additional metrics such as AP scores, as also pointed out in the paper "Promising or elusive? unsupervised object segmentation from real-world single images, NeurIPS 2022".

6) In Section 5 “Related Works”, in the field of object-centric learning, recently, there are a number of works using pretrained feature representations to discovery objects, such as DINOSAUR (Bridging the Gap to Real-World Object-Centric Learning, ICLR 2023), Odin (Object discovery and representation networks, ECCV 2022), and CutLER (Cut and Learn for Unsupervised Object Detection and Instance Segmentation, CVPR 2023). They should be discussed appropriately.

More other related works should be discussed as well, including (1) Invariant Slot Attention: Object Discovery with Slot-Centric Reference Frames, ICML 2023, (2) Spotlight Attention: Robust Object-Centric Learning With a Spatial Locality Prior, arxiv 2023.

To sum up, the paper can be further improved in the following aspects: 1) a better presentation about the technique parts, 2) more concrete experiments to demonstrate better performance in object discovery, 3) discussions about more related and recent works in the field of object-centric learning.

**Questions:**

See above.

---

> ### Author Response · Authors · 2023-11-17
> **Answer to Reviewer fszf**
>
> We would like to express our sincere gratitude for taking the time to provide us with your valuable review and analysis. Your insights are instrumental in helping us further improve our paper.
>
> **Q1. Unsupervised tasks.** We would like to clarify that the image reconstruction task with SLATE, as featured in our study, is indeed an unsupervised task. In this task, the model is given an image and tasked with reconstructing the same image, similar to unsupervised object discovery. This approach encompasses the use of an autoregressive transformer conditioned on slot representations, aimed at reconstructing the input image in latent space.
>
> This methodology is particularly significant as it demonstrates scalability to more complex scenes while maintaining object-level disentanglement. We see the ability to reconstruct an image in latent space, preserving object identities and relationships as one of the key aspects of OCL. Thus, our work, contributes to the field by enhancing the capability of models to handle complex, unsupervised learning scenarios effectively.
>
> We believe that our focus on image reconstruction and set prediction tasks highlights the versatility and applicability of our approach in both supervised and unsupervised learning contexts, thereby underscoring its relevance and importance in the broader landscape of OCL.
>
> **Q2. Combining Contributions (2) and (3).** We agree that Сontributions (2) and (3) should be combined as they both describe the experimental results. We have made this change in the revision of the paper.
>
> **Q3. Use of math symbols instead of English words.** We have made this change in the revision of the paper. Thank you for pointing this out, the introduction of the suggested notation has improved the readability of the paper.
>
> **Q4. Function f_theta(x, u, diag).** Thank you for bringing this ambiguity to our attention. We have clarified the description of the function f_theta(x, u, diag).
>
> **Q5. More complex datasets and additional metrics.** We thank you for your valuable advice and agree that experimenting with more complex datasets and using additional metrics will further improve our paper. Unfortunately, the duration of the rebuttal period does not allow for additional experiments due to the lengthy training of the model. Nevertheless, we can additionally present the results for three seeds for the Shapestacks FG-ARI = 0.87 +- 0.02 dataset and the results of cumputting the AP(0.5) metric for three seeds for the ClevrTEX dataset: AP(0.5) = 0.47 +- 0.15.
>
> **Q6. Section “Related Works".** We have expanded the related works section in the revision of our paper.

---

> > ### Comment · Reviewer_fszf · 2023-12-03
> > **Keep the  rating**
> >
> > Thanks for the feedback of authors, I will keep my negative rating because my primary concern in unsupervised object discovery is not clearly resolved. Nevertheless, I agree that this is a valid extension of SlotAtt, and wouldn't feel upset if it's accepted.

---

### Official Review · Reviewer_7AcU · 2023-10-30

**Soundness:** 4 excellent
**Presentation:** 4 excellent
**Contribution:** 3 good
**Rating:** 8
**Confidence:** 5

**Summary:**

The authors propose a modification of the popular Slot-Attention module in which the slots parameterize a gaussian mixture model, rather than a k-means model (which is how the original SA is typically conceptualized). This allows slots to model both the means and variances (diagonalized covariance) of a latent distribution leading to improved performance on various benchmarks, whilst also allowing "empty" slots to be identified and discarded based on their learned prior mixture weights. The results on various benchmarks are compelling, and the experiments are well-thought out - offering fair comparisons against a vanilla (with implicit gradients) SA baseline, and with some ablations.

**Strengths:**

The paper is very well-written and easy to follow. The methodology is motivated and presented clearly, and the experiments are thorough.

The idea itself is a fairly simple and elegant extension of SA which has not been explored in the community before. As comparisons with SA are made using implicit gradients (which are commonly used for improved training stability of such models now) on various benchmarks, there are good reasons to expect that the proposed SMM model may replace SA in most use-cases (i.e. it improves upon the _practical_ state of the art in a simple way). Given the considerable popularity of SA and its derivatives, this means the paper should have considerable impact in the Object-Centric Learning community.

The ablations compare SA and SMM against their barebones counter-parts (which are closed to traditional k-Means and GMMs), bolstering the suggestion that the learning of the variance is truly beneficial for learning superior slot representations. In addition to comparing on reconstruction and property identification datasets, they show that the quality of edited images (formed through the manipulation of slots) is superior in the SMM model also, lending further credence to the former claim.

**Weaknesses:**

Given the emphasis on the method as an improvement over SA across numerous datasets and kinds of OCL tasks, there are no significant weaknesses in the methodology or choice of experiments. That being said, it would have been interesting to see an analysis of the properties of the learned representations (disentanglement, interpolation/extrapolation over generative factors, etc.), but the paper already contains sufficient information to demonstrate the value of SMMs - most notably the Concept Sampling experiments which show steerability at the level of distinct generative factors.

One minor note is that whilst the authors do take care to fairly compare SA against SMMs, it is still possible that SA with $2D$ slots would have been a "fairer" baseline (though, as they rightly point out, this would have been a model with more learnable parameters), in that the capacity in the slots may have been the limiting factor (unlikely); though I suspect the GMMs would still have performed more strongly (as $D$ is not so much the bottleneck, as the form of the distribution the models can capture). It would also be interesting to see how the behaviour of the two models varied as slots became very small, or sparse regularization was applied to the slot representations.

**Questions:**

Some minor questions which likely reflect ignorance on the part of the reviewer:
* _Redundant Slots_: Do the authors every observe multiple slots sharing the encoding the same object in a scene, or "competition" between the spatial attention over the CNN Feature maps work strongly enough to prevent this, even in SMMs?
* At the end of page one you write that "We believe... set prediction... requires distinguishing objects from each other"- this reads as if you are saying that set prediction is a better measure of OCL competency than object discovery - I am not sure I follow this argument if so, as it seems that entanglement of representations should be less of a hindrance to object discovery than the production of e.g. object-wise masks?
* At the start of section 3 you state that "GRU [...]. takes current and previous slot representations as input and hidden states" but in Algorithm 1 it seems that the GRU is fed only the means?
* It seems as if figure 1 (left) might differ from Algorithm 1 in a few ways (at least, it was sufficiently unclear that perhaps labelling edges would be worthwhile). Most notably, the L2 difference for computing the covariance matrix is taken after the GRU-mean update in the algorithm, but before in the figure. Additionally, I don't think the MLP / LayerNorm are represented in the figure.

Nitpicks:
* The penultimate sentence of the first paragraph in sec 2.1 is somewhat difficult to parse
* The second paragraph in sec 2.1 - "updating iteration as" needs changing
* In the first paragraph of sec 4.3 you describe the role of the 4th channel in the spatial broadcast decoder as being "for the weights of the mixture component" ; whilst this is correct in the context of the sentence, it is slightly confusing given that the slots are represented with mixture components $$\pi$ within the SMM itself. It might be clearer to talk about the weights of _masks_.

---

> ### Author Response · Authors · 2023-11-17
> **Answer to Reviewer 7AcU**
>
> We sincerely appreciate your thoughtful review and in-depth analysis of our paper. Your insights are extremely valuable to us. We have provided detailed explanations in response to your questions, and we hope that these answers effectively address your concerns. We have also taken into account the additional comments you made about our work and corrected them in the revision of our paper.
>
> **Q1. Redundant Slots.** Hypothetically speaking, the SMM should exhibit a reduced tendency for redundant slot assignment compared to the SA model. This is because, in SMM, once a slot has a few vectors assigned, its prior weight decreases, making it less likely to capture information from other objects. In contrast, SA can be conceptualized as having equal prior weights for each slot, which could potentially increase the non-useful flow of information between slots.
>
> However, in our practical observations with the SMM, the situation concerning redundant slots was similar to that in SA. Generally, redundant slots did not capture significant information from the same object. However, there were instances, particularly in the ShapeStacks dataset, where parts of the ground were split between two different slots, a phenomenon we also noted with SA.
>
> **Q2. Competence of set prediction and object detection tasks for OCL.** Our intention was not to suggest that set prediction is a superior measure of OCL competency compared to object discovery. Instead, we aimed to highlight that set prediction is also a valuable and effective indicator for assessing OCL methods. Our point is that set prediction, alongside object discovery, provides a more holistic view of an OCL method's capabilities. It is not about one being better than the other; rather, each task addresses different aspects of OCL proficiency.
>
> **Q3. GRU input.** Thank you for pointing out this inaccuracy in our description of the GRU's input in Section 3. We acknowledge that there is indeed a discrepancy between the text and Algorithm 1. You are correct in pointing out that the GRU is fed only with the mean, not the entire slot representation, which includes both concatenated mean and covariance.
> This oversight in our description has been corrected. We appreciate your attention to detail and your help in improving the clarity and accuracy of our paper.
>
> **Q4. Discrepancy between Figure 1 and Algorithm 1.** Thank you for your insightful feedback regarding Figure 1. We acknowledge the discrepancy you've pointed out, specifically regarding the sequence of the L2 distance computation for the covariance matrix. Algorithm 1 represents the correct order of operations, where this computation occurs post-GRU mean update.
> In the interest of clarity, we chose to simplify Figure 1 by omitting common steps like MLP / LayerNorm to focus on the unique aspects of our model. However, we recognize that this may have caused some confusion. Based on your feedback, we are considering adding these details in a revised figure to align it more closely with the algorithm, ensuring both clarity and completeness.
>
> We appreciate your contribution to enhancing the precision of our manuscript.

---

> > ### Comment · Reviewer_7AcU · 2023-11-22
> > **Response to Clarifications**
> >
> > Thank you for your comprehensive and illuminating response to my questions. The observations regarding background splitting are interesting, and indeed not too surprising given that there is no strong inductive bias toward "background" vs "objects" in SA or SMM (and the spatial competition [as a function of the smoothly varying position embeddings] in slot attention somewhat biases against very large [as a fraction of the receptive field] and varying "things" being captured by a single slot) - though this has been addressed in some works such as uORF.
> >
> > I have no outstanding confusions/questions - I will increase the confidence of my review, but not the scores themselves. I believe this is an excellent paper which makes a worthy contribution by demonstrably improving upon a popular method.

---

### Official Review · Reviewer_HCjc · 2023-10-31

**Soundness:** 3 good
**Presentation:** 2 fair
**Contribution:** 3 good
**Rating:** 6
**Confidence:** 3

**Summary:**

This paper introduces an object-centric architecture that is able to decompose the scene into a set of slots, useful for several downstream tasks ranging from image reconstruction, object discovery and property prediction. While the traditional Slot Attention model generates the slots using a learnable k-means clustering, the proposed method clusters the pixels using a learnable version of the gaussian mixture modeling. Concretely, they are using an iterative approach to estimate not only the centroids of the clusters, but also the covariance associated with each component. The resulting model proves to be beneficial compared to the basic Slot Attention architecture, especially in more difficult scenarios such as low resolution or harder datasets.

**Strengths:**

- The idea of replacing the k-means algorithm with GMM in the slot attention architecture represent an interesting and novel idea. Given that k-means clustering is a particular case of GMM, the resulting method has potential to mimic, and go beyond the capabilities of the slot attention models.
- The method shows improvement on various tasks

**Weaknesses:**

- Since k-means is a particular case of the GMM framework (with the GMM allowing for a learnable variance in clusters), a discussion regarding the advantages of going for the more general model should be included. What are some real world scenarios where a gaussian-based method perform better?
- From the results in Appendix, Table 9 the SMM model seems to be more sensitive to then number of iterations compared to the traditional SA. Is this a consequence of the EM algorithm not converging or are there other optimisation issues that causes this phenomena?
- As mentioning in Section 3, the SMM differs from SA in 3 aspects: the dot product is replaced by the Gaussian density function; both covariance and mean values are updated in the iterative process and the slot representation is a combination of mean and covariance. It would be insightful to see an ablation study showing the contribution of each one of the changes.

**Questions:**

Please see the Weaknesses section

---

> ### Author Response · Authors · 2023-11-17
> **Answer to Reviewer HCjc**
>
> Thank you for your valuable review and analysis. We have responded to your questions and hope that these explanations adequately address your concerns.
>
> **Q1. Advantages of going from k-means to GMM.** As you mentioned, the GMM’s ability to consider variance as part of the final representation is a significant advantage over K-Means. This is particularly important in scenarios where clusters may have the same mean but different variances. Such situations are common in many real-world applications. For example, in image processing, different object categories might have similar color distributions (means) but vary significantly in their spatial spread (variance), which GMM can capture effectively. Moreover, the use of prior weights in GMM provides an additional layer of flexibility and insight. This aspect is especially useful in identifying redundant slots, as we mentioned in our paper.
>
> **Q2. Sensitivity to the number of iterations.** It's indeed typical for GMM to require more iterations for convergence compared to simpler methods like k-means, which may partially explain the observed phenomenon with SMM. However, in our experiments, we didn't find a direct correlation between the inner likelihood and the final performance of the model. This suggests that the increased sensitivity of SMM to iterations may not be solely due to EM algorithm convergence issues, and other factors in the model's optimization process could contribute to this behavior. Further investigation is required to fully understand this phenomenon.
>
> **Q3. Ablation study of the contribution of each change.** Thank you for suggesting an ablation study. While our current study focused on the collective impact of these changes, we agree that isolating each change could provide deeper insights into their respective contributions. An in-depth ablation study was beyond our initial scope, but is a valuable idea for future research. We aim to explore this in subsequent work to better understand the individual and combined effects of these model improvements.

---

> > ### Comment · Reviewer_HCjc · 2023-11-21
> > **Rebuttal reply**
> >
> > Dear Authors,
> >
> > Thank you for addressing my review.
> >
> > I agree with the points mentioned in the Q1 reply. I am still curious if this type of variance difference retains its advantage when applied within a latent space rather than directly on the input. While acknowledging that this topic falls outside the scope of this rebuttal, it would be interesting as a future work to design a synthetic setup in which you can quantify if for example objects with more variety in terms of colour, shape ends up learning a higher variance in the GMM algorithm.
> >
> > Furthermore, I encourage the authors to perform the ablation study mentioned in the review, as it will help the community better understand the necessity of different components, allowing the field to move forward.
> >
> > Overall, I am happy to keep my initial score.
> >
> > Best,
> > Reviewer HCjc

---

### Official Review · Reviewer_spVv · 2023-11-01

**Soundness:** 3 good
**Presentation:** 3 good
**Contribution:** 3 good
**Rating:** 6
**Confidence:** 5

**Summary:**

This paper proposes a generalization of the Slot Attention approach by Locatello et al. 2020, replacing the k-means algorithm with a Gaussian Mixture Model to improve the expressiveness of the slot representations. In Slot Attention (SA), slot representations are cluster centers, which means that SA is limited by the information contained and represented in these cluster centers, whereas SMM represents slots not only as centers of clusters but also incorporate information about the distance between clusters and assigned vectors. Experiment results on standard benchmark datasets showed improved performance over SA.

**Strengths:**

The idea presented in this manuscript is quite sound and intuitive. Overall, the manuscript is well-written and easy to digest. Given the growing interest in Slot Attention, this paper comes timely. It proposes a new direction into how object-centric learning via slot attention could be approached without drastically departing from the main concept while achieving better performance.

**Weaknesses:**

1. There is a typo in Equation 7. The covariance matrix $\Sigma^{\*}$ should be computed based on the updated mean $\mu^{\*}$

2. The qualitative results could be improved, in my opinion. The images depicted in Figure 2 are quite blurry. This makes it quite difficult to assess whether SMM brings actually any substantial improvements over SA or not. I'd recommend the authors to provide higher quality images to strengthen their manuscript.

3. In Figure 2, the images produced from SMM seem quite distorted. These distortions seem more pronounced for the ClevrTex dataset. I am under the impression that given how expressive SMM (when compared to SA) is the reconstructions from the learned slots would be more naturally-looking. It is not clear to me whether the distortions result from the considered Image GPT model, or whether it's because the learned slot representations are not that informative.

4. Comparing *quantitatively* the attention maps learned by SMM against those learned by SA would have been quite helpful. I am under the impression that for more complex image scenes, the attention maps learned by SA would be more accurate than those learned by SMM due to potentially the high-variance involved.

**Questions:**

My main concerns mainly pertain to the quality of the reconstructed images. Any improvements in that aspect would strengthen the manuscript.

---

> ### Author Response · Authors · 2023-11-17
> **Answer to Reviewer spVv**
>
> We appreciate your thorough review and expertise. Below, we have responded to your questions and hope that these answers sufficiently address your concerns.
>
> **Q1. Typo in Equation 7.** Thank you for pointing out this typo. Yes, it should be \mu_{k}^{*} instead of \mu_{k}. We have corrected this error in the text. Unfortunately, LaTeX limitations prevent us from highlighting the formula.
>
> **Q2-Q3. Quality of image reconstruction.** Regarding the blurriness and relatively low resolution of the images (during training we used images with a resolution of 96x96), we acknowledge that it might have impacted the ability to visually assess the improvements our approach brings over the SA. In our paper, we discussed how the limitations of the encoder/decoder models used in our experiments are a significant factor affecting image reconstruction quality. While our quantitative metrics indicated noticeable improvements in the quality of latent space reconstructions by SMM, these enhancements might not be as evident visually in the source space due to these limitations.
>
> We set up training at higher resolutions (128x128 and 256x256), but training at this resolution takes much more time. If the model manages to converge by the end of the rebuttal period, we will include those results in the paper.
>
> **Q4. Comparison of the attention maps quality.** Attention maps in models such as SMM and SA change with each iteration, and typically, only the maps from the final iteration are considered or depicted. However, these final iteration maps might not comprehensively represent the information contained in the slots throughout the iterative process. Therefore, a comparison based solely on these maps may not be fully informative or indicative  of the overall performance of the models.

---

> > ### Comment · Reviewer_spVv · 2023-11-20
> >
> > Thanks for the comments and the updated manuscript. Regarding Q1, yes do provide some samples when available. As for Q4, you are already providing the per-slot attention masks in Figure 4. What I am asking is basically to provide the same for SA so we can have a side-by-side comparison between the two methods.

---

> > > ### Author Response · Authors · 2023-11-23
> > > **Answer to Reviewer spVv**
> > >
> > > **Per-slot attention masks for SMM and SA.** We have included examples of per-slot attention masks for a qualitative comparison of SMM and SA in the Appendix C in Figure 7.
> > >
> > > **Higher resolutions training results.** Unfortunately, the models trained at higher resolutions (128x128 and 256x256) did not converge by the end of the rebuttal period. We believe that no reasonable conclusions can be drawn from the intermediate results; therefore, we have chosen not to include these results. However, we find it important to include the final results and intend to do so in the final version.

---

> > > > ### Comment · Reviewer_spVv · 2023-12-03
> > > >
> > > > Thank you for updating the manuscript and for including the SA masks. My concern has now been addressed. I think your paper deserves to be accepted given the substantial improvements over SA that you bring.

---

### Author Response · Authors · 2023-11-17
**General response to all reviewers**

We thank the reviewers for their dedication and time spent evaluating our paper. We are also sincerely grateful for the thoroughness and depth of the reviews. Taking the comments into consideration, we have uploaded a revised version of the paper and are confident that these changes have enhanced the quality of our work.

---

### Meta-Review · Area_Chair_JAtR · 2023-12-04

**Metareview:**

This paper introduces a simple yet effective change to the Slot Attention model (for unsupervised scene decomposition): instead of following the soft k-Means approach from Slot Attention, the authors parameterize the routing mechanism as a Gaussian mixture model. The effectiveness of this change is validated on a range of compositional / multi-object image datasets.

All reviewers agree that the paper is well-written and that it tackles an important problem. Some of the reviewers especially appreciate the simplicity and elegance of the approach and agree on its novelty.

There are (minor) concerns about the breadth of the experimental validation and the low resolution (64x64) for the presented results. Other concerns about clarity of some parts of the paper were well-addressed in the rebuttal.

I agree with (the majority of) the reviewers that this paper meets the bar for acceptance and that it will be a valuable addition to the growing literature on unsupervised scene decomposition / object-centric learning.

For the final version of the paper, I highly recommend including results for models trained at higher resolution (as discussed in the author/reviewer discussion with Reviewer spVv).

**Justification For Why Not Higher Score:**

Limited breadth of experimental validation (primarily synthetic scenes, monocular images at low resolution).

**Justification For Why Not Lower Score:**

Method is novel, simple/elegant, and of significance. Scientific claims are verified to a sufficient degree.

---

### Decision · Program_Chairs · 2024-01-16

Accept (poster)